# Engineered *Campylobacter jejuni* Cas9 variant with enhanced activity and broader targeting range

Ryoya Nakagawa[1], Soh Ishiguro[2], Sae Okazaki[3], Hideto Mori[4,5], Mamoru Tanaka[2], Hiroyuki Aburatani [6], Nozomu Yachie[2], Hiroshi Nishimasu [1,3,7✉] & Osamu Nureki [1✉]

The RNA-guided DNA endonuclease Cas9 is a versatile genome-editing tool. However, the molecular weight of the commonly used *Streptococcus pyogenes* Cas9 is relatively large. Consequently, its gene cannot be efficiently packaged into an adeno-associated virus vector, thereby limiting its applications for therapeutic genome editing. Here, we biochemically characterized the compact Cas9 from *Campylobacter jejuni* (CjCas9) and found that CjCas9 has a previously unrecognized preference for the $N_3$VRYAC protospacer adjacent motif. We thus rationally engineered a CjCas9 variant (enCjCas9), which exhibits enhanced cleavage activity and a broader targeting range both in vitro and in human cells, as compared with CjCas9. Furthermore, a nickase version of enCjCas9, but not CjCas9, fused with a cytosine deaminase mediated C-to-T conversions in human cells. Overall, our findings expand the CRISPR-Cas toolbox for therapeutic genome engineering.

[1] Department of Biological Sciences, Graduate School of Science, The University of Tokyo, 7-3-1 Hongo, Bunkyo-ku, Tokyo 113-0033, Japan. [2] School of Biomedical Engineering, Faculty of Applied Science and Faculty of Medicine, The University of British Columbia, Vancouver, British Columbia, Canada. [3] Structural Biology Division, Research Center for Advanced Science and Technology, The University of Tokyo, 4-6-1 Komaba, Meguro-ku, Tokyo 153-8904, Japan. [4] Institute for Advanced Biosciences, Keio University, 14-1 Baba-cho, Tsuruoka, Yamagata 997-0035, Japan. [5] Graduate School of Media and Governance, Keio University, 5322 Endo, Fujisawa, Kanagawa 252-0882, Japan. [6] Genome Science Division, Research Center for Advanced Science and Technology, The University of Tokyo, 4-6-1 Komaba, Meguro-ku, Tokyo 153-8904, Japan. [7] Inamori Research Institute for Science, 620 Suiginya-cho, Shimogyo-ku, Kyoto 600-8411, Japan. ✉email: nisimasu@g.ecc.u-tokyo.ac.jp; nureki@bs.s.u-tokyo.ac.jp

The RNA-guided DNA endonuclease Cas9 binds a single-guide RNA (sgRNA) and cleaves double-stranded DNA targets complementary to the RNA guide[1] (Supplementary Fig. 1a), and thus functions in the microbial CRISPR-Cas adaptive immune system[2]. *Streptococcus pyogenes* Cas9 (SpCas9) is the most widely used endonuclease for genome editing in eukaryotic cells[3], and requires an NGG (N = A/T/G/C) protospacer adjacent motif (PAM) downstream of the target sequence for DNA recognition and unwinding[1,4]. Since the catalytically inactive SpCas9 (dSpCas9) serves as an RNA-guided DNA-binding protein, dSpCas9 fusions with various proteins have been applied to new technologies, such as transcriptional regulation, epigenome editing, and chromosomal imaging[5]. In addition, base editors, comprising the SpCas9 D10A nickase (nSpCas9) fused to a cytosine or adenosine deaminase, have enabled C-to-T or A-to-G substitutions at target genomic sites in a guide RNA-dependent manner[6].

*Campylobacter jejuni* Cas9 (CjCas9) is a 984 residue protein and is significantly smaller than SpCas9 (1,368 residues)[7]. Since CjCas9 with an sgRNA can be packaged into an adeno-associated virus vector more efficiently, as compared with SpCas9, the compact CjCas9 is useful for in vivo therapeutic genome editing[8–11]. Previous studies revealed functional and structural differences between CjCas9 and SpCas9[8,12]. While SpCas9 uses sgRNAs with 20-nucleotide (nt) guide lengths, CjCas9 requires 22-nt guide lengths for robust DNA cleavage in human cells[8]. In addition, CjCas9 recognizes the NNNNRYAC (R = A/G; Y = T/C) PAM[8] and the NNNVRYM (V = A/G/C; M = A/C) PAM[12]. While SpCas9 recognizes the NGG PAM in the non-target DNA strand through Arg1333 and Arg1335[13] (Supplementary Fig. 1b), CjCas9 uses a distinct set of amino-acid residues to form base-specific contacts with the PAM nucleotides in both the target and non-target strands[12] (Supplementary Fig. 1c). However, some questions about the enzymatic properties of CjCas9 have remained unanswered. First, the optimal guide lengths for CjCas9 were determined in human cells[8], but not in vitro. Second, the potential preference of CjCas9 for promiscuous PAM sequences has remained uninvestigated. These uncertainties might have hampered the wide use of CjCas9 as a versatile genome-editing tool. In addition, unlike SpCas9[14–16] and *Streptococcus aureus* Cas9[17,18], CjCas9 has not been generally harnessed for base editing, although CjCas9 robustly induced indels in mammalian cells[8].

In this study, we performed the functional characterization and rational engineering of CjCas9. Our biochemical analysis revealed that CjCas9 exhibits robust activity with an optimal 22-nt guide sgRNA and recognizes $N_3VRYAC$ PAMs with unexpected nucleotide preferences. We rationally engineered a CjCas9 variant (enCjCas9), which exhibits enhanced cleavage activity and an expanded targeting scope. The enCjCas9 induced indels at endogenous target sites in human cells more efficiently, as compared with the wild-type CjCas9. Notably, a nickase version of enCjCas9, but not wild-type CjCas9, fused to a cytosine deaminase mediated C-to-T conversions at target sites in human cells. Collectively, our findings substantially enhance the utility of the compact CjCas9 in genome- and base-editing technologies.

## Results

### Biochemical characterization of CjCas9
To examine the optimal guide length for CjCas9, we performed in vitro cleavage experiments, using the purified CjCas9, sgRNAs with 20- to 23-nt guide segments, and linearized plasmid DNA containing a target sequence and the canonical $T_3AACAC$ PAM. CjCas9 with the 20-nt guide sgRNA did not cleave the target DNA (Supplementary Fig. 2a, b). In contrast, CjCas9 with the 21–23-nt guide sgRNAs

efficiently cleaved the target DNA, and the 22-nt guide sgRNA was optimal (Supplementary Fig. 2a, b), consistent with a previous study showing that 22-nt guide sgRNAs are ideal for CjCas9-mediated genome editing in human cells[8].

Since we previously examined the DNA cleavage activities of CjCas9 with a 20-nt guide sgRNA[12], we re-analyzed them using the optimal 22-nt guide sgRNA toward target DNAs with 16 different PAMs, in which the fourth to eighth nucleotides in the canonical $T_3AACAC$ PAM were individually substituted. CjCas9 efficiently cleaved the target DNAs with the $T_3VACAC$ PAMs, but not that with the $T_3TACAC$ PAM (Fig. 1a, Supplementary Fig. 3a–c), confirming the importance of the fourth V in the PAM. CjCas9 cleaved the target DNAs with the $T_3ARCAC$ and $T_3AAYAC$ PAMs, but not those with the $T_3AYCAC$ and $T_3AARAC$ PAMs (Fig. 1a, Supplementary Fig. 3a–c), confirming the requirements of the fifth R and sixth Y for the PAM recognition. CjCas9 cleaved the target DNA with the $T_3AACAC$ PAM more efficiently than that with the $T_3AACBD$ (B = T/G/C; D = A/T/G) PAMs, indicating the importance of the seventh A and eighth C. The crystal structure of the CjCas9–guide RNA–target DNA complex suggested that the seventh A:T and eighth C:G (modeled) pairs in the PAM duplex are recognized by Arg866 through hydrogen bonds with the PAM-complementary T and G nucleobases[12] (Supplementary Fig. 3d), explaining the preference for the seventh A and the eighth C in the PAM. Together, these results revealed that CjCas9 recognizes the $N_3VRYAC$ PAMs, and are essentially consistent with previous studies[8,12].

To examine the potential preference of CjCas9 for specific sequences in the $N_3VRYAC$ PAMs, we measured the DNA cleavage activities of CjCas9 toward 12 different DNA targets encompassing all possible 12 nucleotide combinations at the fourth to sixth positions in the $N_3VRYAC$ PAMs. CjCas9 showed reduced activities toward the $T_3VGYAC$ targets (in particular, the $T_3CGYAC$ targets) relative to the $T_3VAYAC$ targets, although they are included in the $N_3VRYAC$ consensus sequences (Fig. 1b, Supplementary Fig. 3e–g). Collectively, these results indicated that CjCas9 disfavors some combinations in the $N_3VRYAC$ PAMs, such as $N_3RGCAC$ and $N_3CGYAC$.

### Engineering of the enhanced CjCas9 variant
To eliminate the bias of CjCas9-mediated PAM recognition, we sought to engineer a CjCas9 variant with enhanced activity. Previous studies revealed that additional interactions between Cas9 and the nucleic acids improve the DNA cleavage activity[19,20]. We identified 14 residues close to the nucleic acids in the CjCas9–sgRNA–target DNA complex structure[12] (Supplementary Fig. 4a). Accordingly, we introduced mutations that could form new interactions with the guide RNA or the target DNA, purified over 40 CjCas9 variants (20 single mutants and their combinations), and measured their DNA cleavage activities toward the sub-optimal $T_3VGCAC$ targets. Notably, the L58Y and D900K mutations enhanced the DNA cleavage activity, and the L58Y/D900K double mutation further improved the activity of CjCas9 (Supplementary Fig. 4b–f). The crystal structure suggested that Tyr58 (L58Y) and Lys900 (D900K) interact with the guide RNA and the target DNA, respectively (Supplementary Fig. 4a). We will hereafter refer to the L58Y/D900K variant as the enhanced CjCas9 (enCjCas9).

We next compared the in vitro cleavage activities of the wild-type CjCas9 (referred to as CjCas9 for simplicity) and enCjCas9 toward 23 DNA targets with different PAMs. Unlike CjCas9, enCjCas9 efficiently cleaved all of the $T_3VRYAC$ targets, including the sub-optimal $T_3CGYAC$ targets (Fig. 1c, d, Supplementary Fig. 5a–f). Furthermore, enCjCas9 cleaved some non-$N_3VRYAC$

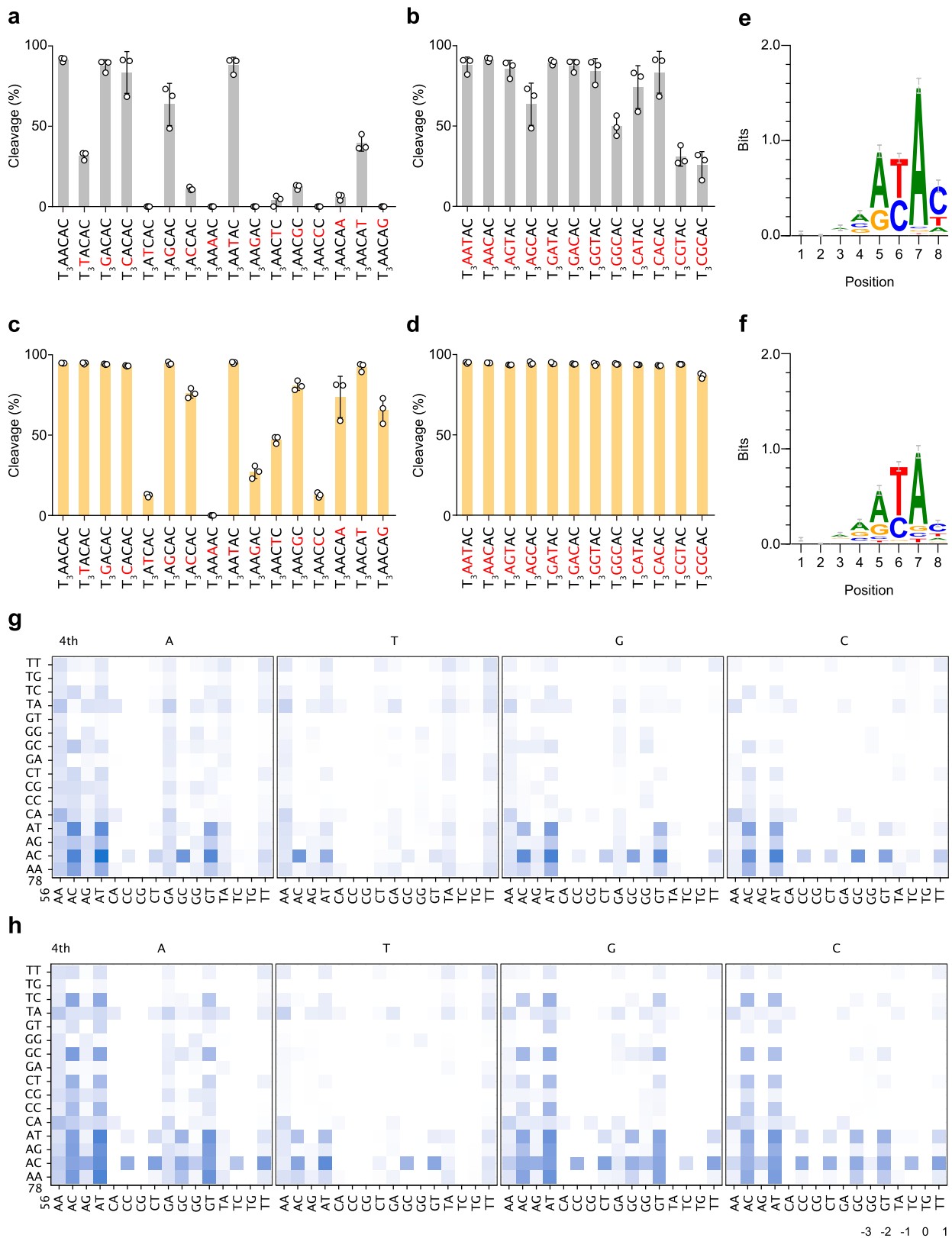

**Fig. 1 In vitro cleavage activities of CjCas9 and enCjCas9. (a–d)** In vitro cleavage activities of CjCas9 (**a**, **b**) and enCjCas9 (**c**, **d**) toward DNA targets with different PAMs. The linearized plasmid targets were incubated with the CjCas9–sgRNA complex at 37 °C for 2 min. Data are mean ± s.d. (*n* = 3). (**e**, **f**) Sequence logos of CjCas9 (**e**) and enCjCas9 (**f**) obtained from the in vitro PAM discovery assay. (**g**, **h**) 2D PAM profiles of CjCas9 (**g**) and enCjCas9 (**h**) obtained from the in vitro PAM discovery assay.

targets, such as T₃TACAC and T₃AACAD (Fig. 1c, Supplementary Fig. 5a–c).

To comprehensively compare the PAM specificities of CjCas9 and enCjCas9, we performed in vitro PAM discovery assays, in which a DNA library containing the target sequence adjacent to a randomized 8-bp sequence was cleaved by the purified CjCas9 (CjCas9 or enCjCas9) with the 22-nt guide sgRNA, followed by deep sequencing of the cleavage products. The sequence logos of the 8-bp random sequences depleted in this assay revealed similar N₃VRYAC PAM sequences for CjCas9 and enCjCas9, although enCjCas9 exhibited more relaxed nucleotide requirements for the fifth, seventh, and eighth PAM positions (Fig. 1e, f). However, although sequence logo representation is widely used to identify functional PAMs, it lacks detailed information about the preferences for individual sequences within promiscuous PAMs. We thus expressed the obtained sequencing data as 2D profiles of the mean $\log_2$ PAM depletion values on all 1,024 sequences at the fourth to eighth PAM positions. The PAM profiles revealed that CjCas9 has preferences among the N₃VRYAC PAMs (Fig. 1g), consistent with our in vitro cleavage data for the individual PAM targets. In contrast, enCjCas9 efficiently recognized the N₃VRYAC PAMs and some non-N₃VRYAC sequences, such as N₃VAYTC and N₃VAYGC (Fig. 1h).

**Genome editing in human cells**. To assess the activities of CjCas9 and enCjCas9 in mammalian cells, we measured indel formation induced by CjCas9 or enCjCas9 at 38 endogenous target sites with the optimal N₃VACAC, sub-optimal N₃VGCAC, and non-N₃VRYAC (N₃TACAC and N₃AACAD) PAMs in the human embryonic kidney (HEK) 293Ta cells. CjCas9 induced indels at the target sites with the optimal N₃VACAC, sub-optimal N₃VGCAC, and non-N₃VRYAC PAMs at 44.0–53.6% (48.5% on average), 10.1–20.7% (16.4% on average), and 3.1–20.8% (12.9% on average) frequencies, respectively (Fig. 2a, b). In contrast, enCjCas9 induced indels at the optimal N₃VACAC, sub-optimal N₃VGCAC, and non-N₃VRYAC PAM sites at 58.4–68.7% (64.8% on average), 20.2–39.8% (32.3% on average), and 14.8–41.0% (29.0% on average) frequencies, respectively (Fig. 2a, b). These results demonstrated that, as compared with CjCas9, enCjCas9 exhibits higher cleavage activities and broader targeting ranges in human cells, consistent with our in vitro cleavage data. Next, we compared the genome-editing efficiencies of CjCas9 and enCjCas9 with those of SpCas9 at ten target sites with NGGAACAC PAMs, which can be accessed by both CjCas9 (N₃VRYAC PAM) and SpCas9 (NGG PAM). CjCas9, enCjCas9, and SpCas9 generated indels at these ten sites with 25.0–86.6% (60.8% on average), 49.0–87.5% (74.3% on average), and 43.8–75.4% (61.6% on average) frequencies, respectively (Fig. 2c). These results indicated that CjCas9 with optimal 22-nt guide sgRNAs can induce indels at target sites with appropriate PAMs, at efficiencies comparable to or higher than those of SpCas9, as previously reported[8]. To examine the specificities of CjCas9 and enCjCas9, we measured their editing activities toward two on-target sites (AAVS1-TS1 and AAVS1-TS8) and six off-target sites (AAVS1-TS1-02–04 and AAVS1-TS8-02–04), which were previously identified by the Digenome-seq method[8]. enCjCas9 edited the two on-target sites, but not the six off-target sites, more efficiently than CjCas9 (Supplementary Fig. 6), indicating that the specificity of enCjCas9 is comparable to that of CjCas9.

**Base editing in human cells**. Target-AID, comprising the SpCas9 D10A nickase mutant fused to the *Petromyzon marinus* cytosine deaminase 1 (PmCDA1) and the uracil DNA glycosylase inhibitor (UGI), mediates C-to-T conversion at target genomic sites[15]. We replaced the SpCas9 D10A nickase in Target-AID (referred to as

SpCas9-AID for comparison) with the CjCas9 D8A and enCjCas9 D8A nickases to create CjCas9-AID and enCjCas9-AID, respectively. We examined whether CjCas9-AID and enCjCas9-AID could mediate C-to-T conversions at 38 endogenous target sites (identical to those tested for indel formation) in HEK293 cells. Unexpectedly, CjCas9-AID did not induce C-to-T conversions at the tested target sites (Fig. 3a, b). In contrast, enCjCas9-AID induced C-to-T conversions at 20 target sites at >5% frequencies (the optimal N₃VACAC, sub-optimal N₃VGCAC, and non-N₃VRYAC sites at 15.8–25.1% (21.5% on average), 4.3–13.2% (8.0% on average), and 1.2–12.4% (6.2% on average) frequencies, respectively) (Fig. 3a, b). We also compared the base-editing efficiencies of CjCas9-AID, enCjCas9-AID, and SpCas9-AID at the ten target sites with the NGGAACAC PAMs. enCjCas9-AID induced C-to-T conversions at most of the target sites, albeit at lower efficiencies than those of SpCas9-AID (Fig. 3c). CjCas9-AID induced C-to-T conversions at some of the NGGAACAC sites (in particular, the CASZ1 site), although the reason is unknown. enCjCas9-AID induced C-to-T conversions predominantly at the −21 to −8 positions in target sites (Supplementary Fig. 7). These data revealed that enCjCas9, but not CjCas9, can be harnessed for base editing technologies.

## Discussion

In this report, we biochemically characterized the compact CjCas9, and found that CjCas9 efficiently cleaves a target DNA with a 22-nt guide sgRNA, consistent with a previous study in human cells[8]. In contrast, CjCas9 failed to cleave a target DNA with a 20-nt guide sgRNA, while SpCas9 exhibited robust DNA cleavage activity with 20-nt guide sgRNAs. These results reinforce the concept that the optimal guide lengths are different among Cas9 enzymes, as previously reported[8,21,22]. A structural comparison of the Cas9–guide RNA–target DNA complexes suggested that these functional differences are related to structural differences in their REC domains, which interact with the 5′ end of their guide RNAs[22], although the precise mechanisms remain unknown. We also found that CjCas9 has a previously unrecognized bias for N₃VRYAC PAMs, and reduced activities toward target sites with sub-optimal PAMs, such as N₃VGYAC. Although CjCas9 exhibited similar PAM preference trends in vitro and in human cells, the genome- and base-editing activities of CjCas9 were highly dependent on the target loci. For example, CjCas9 edited the DYRK1A and MECP2 sites with the N₃GACAC PAMs at >80% and <10% frequencies, respectively, suggesting that its editing efficiencies are substantially affected by the genomic context, as observed previously[8]. These results highlight the importance of in vitro cleavage experiments for the accurate determination of the PAM specificity, as described previously[23–25]. In addition, our results demonstrated that a 2D PAM profile[24,25] or a PAM wheel[26], rather than a 1D sequence logo, is critical to fully visualize functional PAM sequences, particularly when Cas9 recognizes relatively long, promiscuous PAMs. Collectively, our findings underscore the practical importance of the use of optimal 22-nt guide sgRNAs and the selection of appropriate PAMs in CjCas9-mediated genome engineering applications.

Based on the structural information, we introduced the L58Y and D900K mutations into CjCas9 to create the enCjCas9 variant with enhanced activity. The crystal structure of CjCas9[12] suggests that Lys900 (D900K) interacts with the backbone phosphate of the target DNA, as observed in the engineered Cas9 variants, such as FnCas9-RHA[19] and SpCas9-NG[20]. In contrast, Tyr58 (L58Y) apparently forms a stacking interaction with U48 in the guide RNA, which is similar to that between the equivalent Tyr72 and U50 in the SpCas9 structure[27], indicating that additional

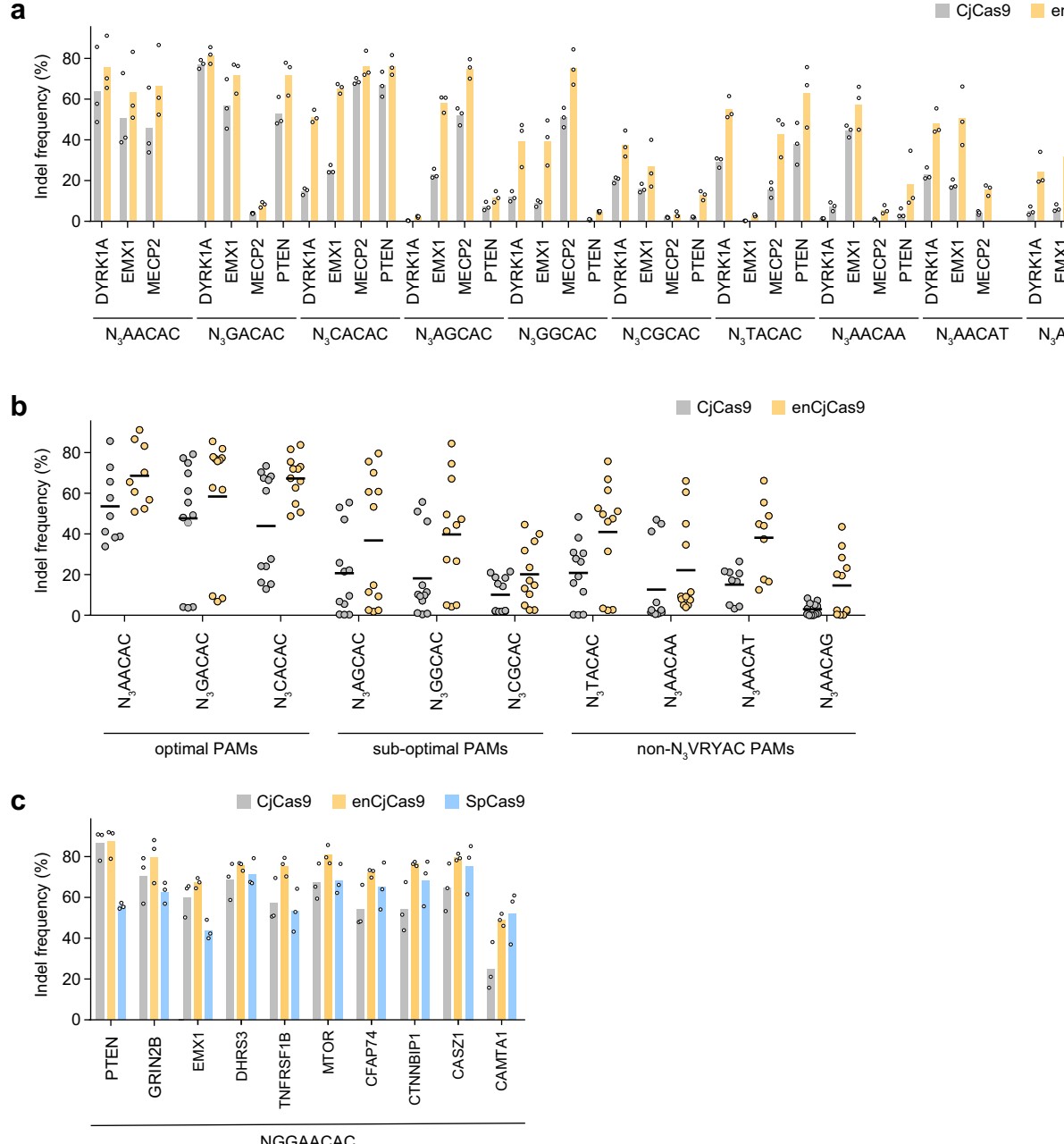

**Fig. 2 Genome editing by CjCas9 and enCjCas9. (a)** Indel formation by CjCas9 (gray) and enCjCas9 (orange) at endogenous target sites in HEK293Ta cells ($n = 3$). (**b**) Summary of the genome-editing efficiencies of CjCas9 (gray) and enCjCas9 (orange) in (**a**). Bars indicate medians. (**c**) Indel formation by CjCas9 (gray), enCjCas9 (orange), and SpCas9 (blue) at endogenous target sites in HEK293Ta cells ($n = 3$).

interactions with the guide RNA can also contribute to facilitating Cas9-mediated DNA cleavage. These results suggest that the reinforcement of protein–nucleic-acid interactions is a widely applicable and effective strategy to engineer CRISPR-Cas enzymes with enhanced activity.

We found that CjCas9 edits target genomic sites with optimal $N_3VRYAC$ PAMs as efficiently as SpCas9, as previously reported[8]. In contrast, CjCas9-AID mostly failed to induce C-to-T conversions even at the identical target sites, consistent with the fact that CjCas9-based cytosine base editors are not currently available, although a CjCas9-based adenosine base editor was reported recently[28]. Notably, we determined that enCjCas9-AID, unlike CjCas9-AID, can induce C-to-T conversions at target sites in human cells. It is possible that the compact enCjCas9-AID (1,437 residues, 4.3 kb) can be

packaged into an adeno-associated virus vector, thereby facilitating in vivo base editing. It will be important to examine whether the enCjCas9 nickase can be combined with other cytosine deaminases[14,17,18] and adenosine deaminases[16]. Furthermore, the catalytically inactive enCjCas9 could serve as a compact RNA-guided DNA-binding platform applicable to other technologies, such as transcriptional regulation.

## Methods

**Sample preparation**. The CjCas9 proteins were prepared, as described previously[12]. Briefly, *Escherichia coli* Rosetta 2 (DE3) cells (Novagen) were transformed with the modified pE-SUMO-CjCas9 vector[12], and then cultured at 37 °C in LB medium (containing 20 mg/L kanamycin) until the $OD_{600}$ reached 0.8. After the addition of 0.1 mM isopropyl β-D-thiogalactopyranoside (Nacalai Tesque), the *E. coli* cells were incubated at 20 °C for 20 h, and then harvested by centrifugation.

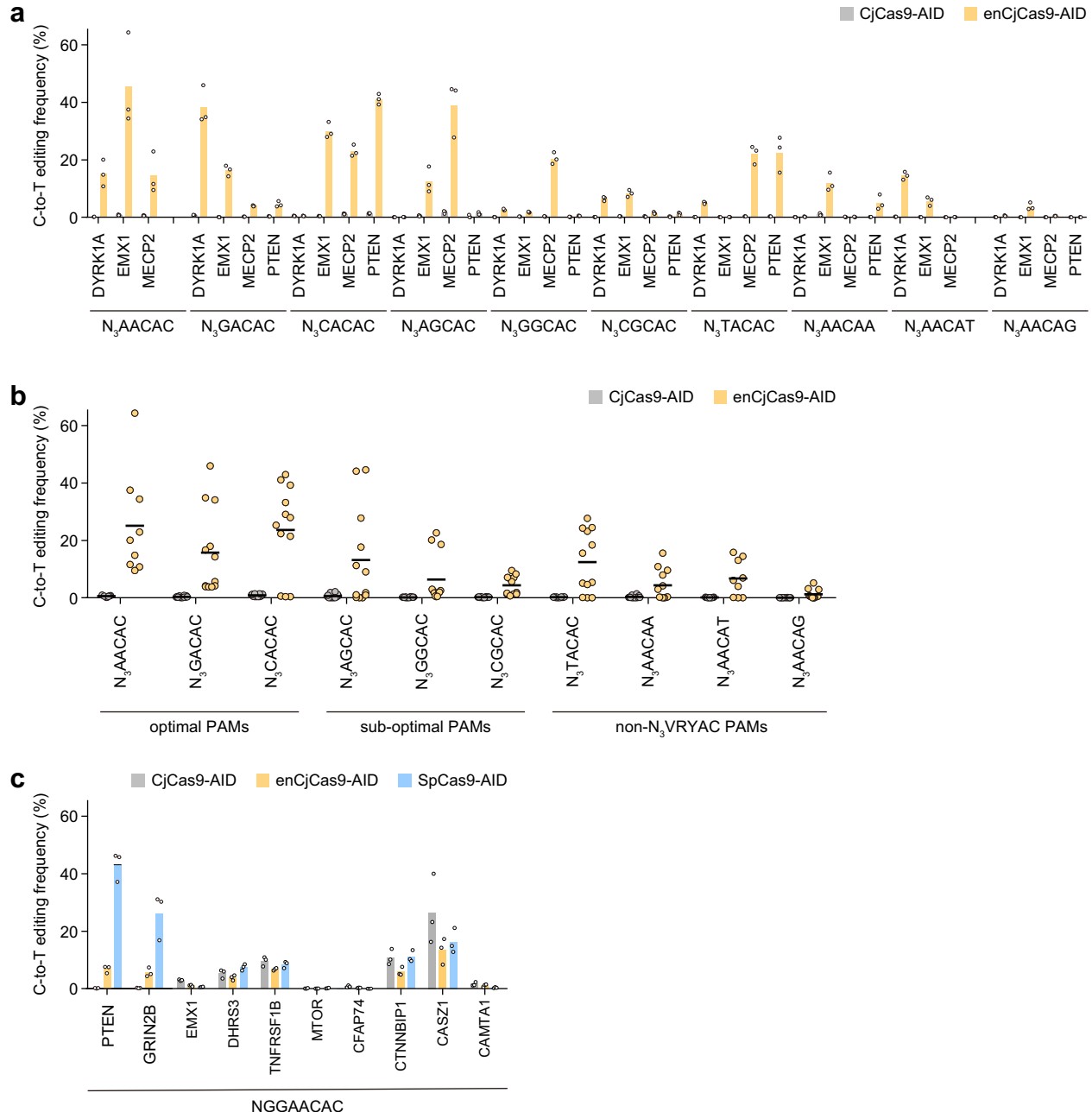

**Fig. 3 Base editing by CjCas9-AID and enCjCas9-AID. (a)** C-to-T conversion by CjCas9-AID (gray) and enCjCas9-AID (orange) at endogenous target sites in HEK293Ta cells ($n = 3$). (**b**) Summary of the base-editing efficiencies of CjCas9-AID (gray) and enCjCas9-AID (orange) in (**a**). Bars indicate medians. (**c**) C-to-T conversion by CjCas9-AID (gray), enCjCas9-AID (orange), and SpCas9-AID (blue) at endogenous target sites in HEK293Ta cells ($n = 3$).

The *E. coli* cells were resuspended in buffer A (20 mM Tris-HCl, pH 8.0, 20 mM imidazole, and 1 M NaCl), lysed by sonication, and then centrifuged. The supernatant was mixed with Ni-NTA Superflow resin (QIAGEN), and the mixture was loaded into a Poly-Prep Column (Bio-Rad). The protein was eluted with buffer B (20 mM Tris-HCl, pH 8.0, 0.3 M imidazole, and 0.3 M NaCl). To remove the His$_6$-SUMO-tag, the purified protein was mixed with SUMO protease and dialyzed at 4 °C for 2 h against buffer C (20 mM Tris-HCl, pH 8.0, 40 mM imidazole, and 0.3 M NaCl). The protein was passed through the Ni-NTA column equilibrated with buffer C. The protein was loaded onto a HiTrap Heparin HP column (GE Healthcare) equilibrated with buffer D (20 mM Tris-HCl, pH 8.0, and 0.3 M NaCl). The protein was eluted with a linear gradient of 0.3–2 M NaCl. The purified proteins were stored at −80 °C until use. The mutations were introduced by an inverse PCR-based method, and the sequences were confirmed by DNA sequencing. The sgRNA was transcribed in vitro with T7 RNA polymerase, using a partially double-stranded DNA template (Supplementary Tables 1 and 2). The transcribed RNA was purified by 8% denaturing (7 M urea) polyacrylamide gel electrophoresis.

**In vitro cleavage experiments**. First, in vitro cleavage experiments were performed, using the purified CjCas9, four sgRNAs with different guide lengths (20–23-nt), and a linearized pUC plasmid containing the target sequence (complementary to the 20–23-nt guides) with the T$_3$AACAC PAM (Supplementary Table 1). Next, the cleavage activities CjCas9 were measured, using the optimal 22-nt guide sgRNA and linearized pUC plasmids with 16 different PAMs. The linearized plasmid DNA (100 ng, 4.7 nM) was incubated at 37 °C for 0.5–5 min with the CjCas9–sgRNA complex (50 nM) in 10 μL of reaction buffer, containing 20 mM HEPES, pH 7.5, 100 mM KCl, 2 mM MgCl$_2$, 1 mM DTT, and 5% glycerol. The reaction was stopped by the addition of quench buffer, containing EDTA (20 mM final concentration) and Proteinase K (40 ng). The reaction products were resolved, visualized, and quantified with a MultiNA microchip electrophoresis device (SHIMADZU).

**PAM discovery assay**. The PAM discovery assays were performed essentially as previously described[20], using a library of pUC119 plasmids containing the target sequence identical to that used in in vitro cleavage experiments and an 8-bp

randomized sequence downstream of the target sequence. The plasmid library was prepared by inserting a 100-bp DNA fragment (IDT) containing the 22-nt target sequence and an 8-bp randomized sequence into a pUC119 plasmid, using Gibson Assembly (NEB). The plasmid library was incubated at 37 °C for 5 min with the CjCas9–sgRNA (22-nt guide) complex (50 nM), in 50 μL of reaction buffer. The reactions were quenched by the addition of Proteinase K, and then purified using a Wizard DNA Clean-Up System (Promega). The purified DNA samples were amplified for 25 cycles, using primers containing common adapter sequences (Supplementary Table 2). After column purification, each PCR product (~5 ng) was subjected to a second round of PCR for 15 cycles, to add custom Illumina TruSeq adapters and sample indices. The sequencing libraries were quantified by qPCR (KAPA Biosystems), and then subjected to paired-end sequencing on a MiSeq sequencer (Illumina) with 20% PhiX spike-in (Illumina). The sequencing reads were demultiplexed by primer sequences and sample indices, using NCBI Blast + (version 2.8.1) with the blastn-short option. For each sequencing sample, the number of reads for every possible 8-nt PAM sequence pattern ($4^8 = 65,536$ patterns in total) was counted and normalized by the total number of reads in each sample. For a given PAM sequence, the enrichment score was calculated as $\log_2$-fold enrichment as compared to the untreated sample. PAM sequences with enrichment scores of −2.0 or less were used to generate the sequence logo representation, using WebLogo (version 3.7.1)[29]. The cumulative distribution and histogram of the read count of each PAM in the unedited sample confirmed that the plasmid library has sufficient coverage for the individual PAM sequences.

**Genome- and base-editing analyses in human cells**. Genome- and base-editing analyses were performed in triplicate, according to the protocol described previously[30]. Briefly, HEK293Ta cells were maintained in DMEM (Sigma) supplemented with 10% (v/v) fetal bovine serum (FBS) (Thermo Fisher Scientific) and 1% Penicillin-Streptomycin (Sigma), at 37 °C in a 0.05% $CO_2$ atmosphere. HEK239Ta cells were seeded at $5×10^3$ cells per well in collagen I-coated 96-well plates, 24 h prior to transfection. HEK239Ta cells were transfected with a CjCas9 plasmid or a CjCas9-derived base-editor plasmid (120 ng) and an sgRNA plasmid (40 ng), using Polyethylenimine Max (Polysciences) (1 mg/mL, 0.5 μL) in PBS (50 μL) (Supplementary Table 3). The cells were harvested 3 days after transfection, treated with 50 mM NaOH (100 μL), incubated at 95 °C for 10 min, and then neutralized with 1 M Tris-HCl, pH 8.0 (10 μL). The obtained genomic DNA was subjected to two rounds of PCR, to prepare the library for high-throughput amplicon sequencing. Genomic regions targeted by sgRNAs were PCR-amplified to add custom primer-landing sequences (Supplementary Tables 2 and 4). The PCR products were purified by AMPure XP magnetic beads (Agencourt), and then subjected to a second round of PCR to attach the custom Illumina TruSeq adapters with sample indices. After size-selection by agarose gel electrophoresis and column purification, the sequencing libraries were quantified using a KAPA Library Quantification Kit Illumina (KAPA Biosystems), multiplexed, and subjected to paired-end sequencing (600 cycles), using a MiSeq sequencer (Illumina) with 20% PhiX spike-in (Illumina). The sequencing reads were demultiplexed, based on sample indices and primer sequences. Using NCBI BLAST + (version 2.6.0) with the blastn-short option, the sequencing reads were mapped to the reference sequences to identify indels and substitutions in the target regions. To remove common PCR errors and somatic mutations, we removed sequencing reads containing mutations (>1% frequency) commonly observed in the control samples from the edited samples, and then normalized the editing frequencies for the target sites by subtracting the mutation frequencies of the control samples from those of the edited samples.

**Statistics and Reproducibility**. In vitro cleavage experiments were performed at least three times. Data are shown as mean ± s.d. ($n = 3$). Kinetics data were fitted with a one-phase exponential association curve, using Prism (GraphPad).

**Reporting summary**. Further information on research design is available in the Nature Research Reporting Summary linked to this article.

## Data availability

The NGS data have been deposited in the NCBI Sequence Read Archive (SRA) under accession codes PRJNA795197. The plasmids used for in vivo editing experiments can be accessed in Addgene under accession codes 180762, 180763, 180764, 180765, and 180767. Any remaining information can be obtained from the corresponding author upon reasonable request.

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

## Acknowledgements

This work was supported by JSPS KAKENHI Grant Numbers 18H02384 (H.N.), 18K19284 (H.N.), and Inamori Research Institute for Science (H.N.), and AMED Grant Number JP19am0401005 (N.Y., H.N., and O.N.).

## Author contributions

R.N. performed biochemical experiments with assistance from S.O. and H.N.; S.I., H.M., M.T., H.A., and N.Y. performed cell biological experiments; R.N. and H.N. wrote the manuscript with assistance from S.I., H.M., N.Y., and O.N.; H.N. and O.N. supervised all of the research.

## Competing interests

The authors declare the following competing interests: O.N. is a co-founder, board member, and scientific advisor for Modalis and Curreio. The remaining authors declare no competing interests.
