## [Peer Review File · Communications Biology]

Reviewers' comments:

Reviewer #1 (Remarks to the Author):

Comments for the Author

CRISPR genome editing technology is versatile and efficient and has revolutionized biological research. SpCas9 has become the most popular and most characterized CRISPR tool but other CRISPR variants and orthologs have also been either created or characterized. Nakagawa et al. reports on the characterization and further improvements of such an ortholog, *Campylobacter jejuni* Cas9 (CjCas9). They determined the optimal spacer lengths of CjCas9 (22mer) in vitro, refined its PAM preferences (N3VRYAC) and developed an enhanced CjCas9 variant, the nickase version of which can be exploited for base editing.

The most interesting achievement of this study is the development of the enhanced CjCas9 variant, which apparently has higher activity than the WT CjCas9 and it may become a useful alternative in genome engineering applications.

The weaknesses of the study are the confusing presentation of the data and overly strong statements about the PAM specificity using only a single protospacer (in vitro and PAM discovery assay) for all PAM sequences examined or three target sequences per PAMs (in cellulose experiments). This may be suitable for discerning the major features of the PAM preferences but not for fine mapping of it, due to the known target sequence dependencies of the PAM preferences of the Cas nucleases. Furthermore, to appreciate the usefulness of enCjCas9, it would be interesting to see its specificity in relation to CjCas9.

Specific comments:

1. The actual spacer length used in the in vitro experiments is not clear, please clarify: "The pUC119 plasmid, containing the 23-nt target sequence and the PAMs, was used as the 253 substrate for in vitro cleavage experiments"

"Since we previously examined the DNA cleavage activities of CjCas9 with a 20-nt guide 83 sgRNA, we re-analyzed them using the optimal 22-nt guide sgRNA toward target DNAs with 84 16 different PAMs,"

which statement is correct?

2. "The 55 optimal guide lengths for CjCas9 were determined in human cells but not in vitro." I wonder, if the authors have any rationale or citation from the literature which states that the optimal spacer length of a Cas nuclease differs between in vitro and in human cells.

3. "To remove common PCR

302 errors and somatic mutations, sequencing reads with the same mutations observed in the control

303 samples (null vector transfection) were removed, and then the mutation frequencies of the target

304 sites were normalized by subtracting those of the same mutations observed in the control 305 samples."

Please clarify whether they were removed, or control reads were subtracted from them?

4. Supplementary fig 4 is not referred to in the MS.

5. "We purified more than 40 CjCas9 114 variants and measured their cleavage activities toward the sub-optimal T3VGCAC targets 115 (Supplementary Fig. 5b)."

Neither Sup Fig 5b or 4b, nor any figures of the MS contain these data. It would be interesting to really see which positions and substitutions were tested and how the other 38 mutations affected activity.

6. in Sup Fig 6., the positions labelled are in the protospacer and not in the spacer as stated.

7. As far as I managed to understand, the authors used the same protospacer sequence in the PAM discovery assay and in the in vitro experiments. This sequence should be stated everywhere in the fig legends.

The author did not mention, but because they are using plural ("The 270 sequencing libraries were quantified by qPCR") they likely did parallel samples. This should be clarified, and more details should be provided of the libraries, such as the coverage for the individual PAM sequences, the read distribution, what the minimal read number was in the initial library, etc.

8. They should upload the NGS data to public repository.

9. No information provided for library preparation, I could not figure out how they used the reverse and forward primers in sup table 2 to generate a library with randomized 8 nucleotides.

10. No information provided for the generation of the mutant DNA constructs, neither for the

enCjCas9 variants nor for any of the 40 mutant variants, (except that "The mutations 246 were introduced by a PCR-based method");

No oligoes/primers are provided for them; the only primer pairs that are provided for L58Y and for D900K themselves are not suitable in my estimation for generating the mutations.

11. What is the percentage of the SDS gel on sup Fig 4.?

12. Do the authors have any explanations why CjCas9 did not work as base editor on targets that are cleaved efficiently by it?

Reviewer #2 (Remarks to the Author):

Following up an earlier publication that characterized CjCas9, Nakagawa et al. re-analyzed PAM specificity using in vitro cleavage with a longer guide sequence (22mer) that previously used (20mer). The PAM is confirmed to be T3VRYAC. Authors then made an important effort in examining the capability of CjCas9 in genome and base-editing in human cells. A mutant variant, L58/K900 (enCjCas9), was also constructed based on its improved activity although with a slightly relaxed requirement for PAM position 7. In human cells, HEK293, CjCas9 and enCjCas9 showed high efficiency in inducing indels and base editing. This effort is a welcome advancement in genome editing field as CjCas9 is the smallest Cas9, which can improve portability. There are significant amount of data presented that support the conclusions. There are some suggestions that can improve the quality of the paper:

1) The efficiency demonstrated by CjCas9 (both wild-type and enhanced) in human cells is quite high and close to or better than that of SpCas9. Thus, this result is significant as many smaller and Type II-C Cas9 show decreased in vitro cleavage activities. It perhaps is important for authors to discuss the process they used to optimize genome editing with CjCas9. Was there a major step/procedure that significantly improved the behavior of CjCas9 in HEK293 cells?

2) The statement in discussion (Page 7, line 195) suggests that the target loci dependent activity of CjCas9 is a result of genome context. However, it appears that different targets are also associated with different PAM. For instance, the worst targets seem to be with N3AACAG (Figure 2a) rather than genome location. Thus, can authors re-qualify what they meant?

3) Page 5, line 50, The "V" designation is incorrect, it should be A/C/G not A/T/G. This created some confusion on description of the PAM specificities;

4) Supplementary Figure 5 does not show the 40 CjCas9 variants as described. Furthermore, can authors explain their rationale for these 40 variants?

5) There does not seem to have any information on the CjCas9 or CjCas9-base editor plasmids used for HEK293 transfection.

Reviewer #3 (Remarks to the Author):

The authors first determined the optimal guide RNA length by using an in vitro cleavage assay at a target sequence. Then, the authors also found the optimal PAM with another in vitro cleavage assay, again using only one target sequence. The results of both of these experiments are essentially in line with previous studies. Next, they developed enCjCas9, a variant of CjCas9, by introducing two mutations, L58Y and D900K. An in vitro cleavage assay showed that enCjCas9 has broader PAM compatibility than CjCas9. Then, the authors compared enCjCas9 with CjCas9 and SpCas9 in human cells. EnCjCas9 showed higher efficiencies than CjCas9 or SpCas9, although only two sites were used for the comparison with SpCas9. Next, the authors generated cytosine base editor by adding deaminase (PmCDA1) and UGI to CjCas9 D8A and enCjCas9 D8A nickases. Base editing was achieved only by enCjCas9 but its efficiencies were substantially lower than that of the SpCas9 nickase-based base editor.

Development of enCaCas9, a small and efficient Cas9 nuclease, would be useful to the field by extending the range of possible genome editing tools. However, there are several points that should be addressed before this manuscript can be published.

Major comments

1. The authors should test off-target effects of enCjCas9 using a minimum of two, or preferably three (or more), on-target sequences. We recommend using unbiased methods such as GUIDE-seq.
2. Only two target sequences were used to compare the efficiencies of enCjCas9 and SpCas9. We recommend that the authors conduct the same experiments using more target sequences (preferably more than five) to draw more generalized conclusions supported by more compelling evidence.
3. Can the enCjCas9-base editor sequence be packaged into one AAV vector? If not, the impact of enCjCas9 would be somewhat limited.

Minor comments

1. "We purified more than 40 CjCas9 variants, and measured their cleavage activities toward the sub-optimal T3VGCAC targets (Supplementary Fig. 5b). Notably, the L58Y and D900K mutations enhanced the DNA cleavage activity, and the L58Y/D900K double mutation further improved the activity of CjCas9 (Supplementary Fig. 5c-f)."
Wrong figure numbers.
2. Line 146, there is a typo: sup-optimal

Reviewer #1:

CRISPR genome editing technology is versatile and efficient and has revolutionized biological research. SpCas9 has become the most popular and most characterized CRISPR tool but other CRISPR variants and orthologs have also been either created or characterized. Nakagawa et al. reports on the characterization and further improvements of such an ortholog, Campylobacter jejuni Cas9 (CjCas9). They determined the optimal spacer lengths of CjCas9 (22mer) in vitro, refined its PAM preferences (N3VRYAC) and developed an enhanced CjCas9 variant, the nickase version of which can be exploited for base editing. The most interesting achievement of this study is the development of the enhanced CjCas9 variant, which apparently has higher activity than the WT CjCas9 and it may become a useful alternative in genome engineering applications. The weaknesses of the study are the confusing presentation of the data and overly strong statements about the PAM specificity using only a single protospacer (in vitro and PAM discovery assay) for all PAM sequences examined or three target sequences per PAMs (in cellulo experiments). This may be suitable for discerning the major features of the PAM preferences but not for fine mapping of it, due to the known target sequence dependencies of the PAM preferences of the Cas nucleases. Furthermore, to appreciate the usefulness of enCjCas9, it would be interesting to see its specificity in relation to CjCas9.

We thank the reviewer for the positive comments.

COMMENTS:

1. *The actual spacer length used in the in vitro experiments is not clear, please clarify:
"The pUC119 plasmid, containing the 23-nt target sequence and the PAMs, was used as the substrate for in vitro cleavage experiments" "Since we previously examined the DNA cleavage activities of CjCas9 with a 20-nt guide sgRNA, we re-analyzed them using the optimal 22-nt guide sgRNA toward target DNAs with 16 different PAMs,"
which statement is correct?*

First, we performed *in vitro* DNA cleavage experiments, using CjCas9, four sgRNAs with different guide (spacer) lengths (20–23-nt), and a linearized plasmid DNA containing the 23-nt target sequence (complementary to all of the 20–23-nt guides) with the T₃AACAC PAM, and confirmed

that the sgRNA with the 22-nt guide segment is optimal for CjCas9-mediated DNA cleavage. Using the optimal 22-nt guide sgRNA, we then measured the cleavage activities CjCas9 toward target DNAs with 16 different PAMs. We added these explanations in the revised manuscript.

2. *The optimal guide lengths for CjCas9 were determined in human cells but not in vitro." I wonder, if the authors have any rationale or citation from the literature which states that the optimal spacer length of a Cas nuclease differs between in vitro and in human cells.*

We consider that the optimal spacer length for a Cas nuclease is essentially the same *in vitro* and in human cells. Indeed, our *in vitro* DNA cleavage experiments revealed that the 22-nt guide length was optimal for CjCas9, consistent with a previous study showing that 22-nt guide lengths are optimal for CjCas9-mediated genome editing in human cells (Kim *et al.*, *Nat. Commun.* 2017). Nonetheless, it is important to determine the optimal guide length by *in vitro* DNA cleavage experiments, which allow accurate measurements of Cas9-catalyzed DNA cleavage, as compared to *in vivo* genome-editing experiments, which evaluate indel formation after Cas9-induced double-strand breaks and thus could be affected by many factors, such as genomic contexts. Thus, we determined the optimal guide length for CjCas9 by *in vitro* cleavage experiments in this study.

3. *"To remove common PCR errors and somatic mutations, sequencing reads with the same mutations observed in the control samples (null vector transfection) were removed, and then the mutation frequencies of the target sites were normalized by subtracting those of the same mutations observed in the control samples." Please clarify whether they were removed, or control reads were subtracted from them.*

To remove common PCR errors and somatic mutations, we removed sequencing reads containing the mutations (>1% frequency) commonly observed in the control samples from the edited samples, and then normalized the editing frequencies for the target sites by subtracting the mutation frequencies for the control samples from those for the edited samples. We have added the detailed information in the revised manuscript.

4. *Supplementary fig 4 is not referred to in the MS.*

Thank you for pointing it out. We have referred to Supplementary Figure 4 in the revised manuscript.

5. *We purified more than 40 CjCas9 variants and measured their cleavage activities toward the sub-optimal T3VGCAC targets (Supplementary Fig. 5b)." Neither Sup Fig 5b or 4b, nor any figures of the MS contain these data. It would be interesting to really see which positions and substitutions were tested and how the other 38 mutations affected activity.*

Thank you for the helpful comment. We identified 14 residues close to the bound nucleic acids in the CjCas9–sgRNA–DNA complex structure. Thus, we purified over 40 variants (20 single mutants and their combinations), and measured their DNA cleavage activities *in vitro*. Since only two mutations (L58Y and D900K) substantially improved the DNA cleavage activity of CjCas9, we focused on the L58Y and D900K mutations in the original manuscript for clarity. According to the reviewer’s suggestion, we have added a brief statement about the other mutations and showed the 14 mutated residues in Supplementary Figure 4a.

6. *in Sup Fig 6., the positions labelled are in the protospacer and not in the spacer as stated*

We changed the “spacer” to “protospacer” in the new Supplementary Figure 7 (the previous Supplementary Figure 6).

7. *As far as I managed to understand, the authors used the same protospacer sequence in the PAM discovery assay and in the in vitro experiments. This sequence should be stated everywhere in the fig legends. The author did not mention, but because they are using plural ("The 270 sequencing libraries were quantified by qPCR") they likely did parallel samples. This should be clarified, and more details should be provided of the libraries, such as the coverage for the individual PAM sequences, the read distribution, what the minimal read number was in the initial library, etc.*

Thank you for the helpful comment. For the PAM discovery assays, we used a single plasmid library containing the 22-nt protospacer (target) sequence identical to that used in the *in vitro* cleavage experiments. The cumulative distribution and histogram of the read count of each PAM in

the unedited sample revealed that our library has sufficient coverage for the individual PAM sequences (Figure L1). According to the reviewer's suggestion, we have added more detailed information about the library in the revised manuscript.

Figure L1. Cumulative distribution and histogram of the plasmid library used for the PAM discovery assays.

8. *They should upload the NGS data to public repository.*

According to the reviewer's suggestion, we are going to upload the NGS data into the NCBI Sequence Read Archive (SRA) before the manuscript is accepted.

9. *No information provided for library preparation, I could not figure out how they used the reverse and forward primers in sup table 2 to generate a library with randomized 8 nucleotide.*

We have added more detailed information about the library preparation in the revised manuscript.

10. *No information provided for the generation of the mutant DNA constructs, neither for the enCjCas9 variants nor for any of the 40 mutant variants, (except that "The mutations were introduced by a PCR-based method")*

We have added more details about the mutant preparation in the revised manuscript.

11. *What is the percentage of the SDS gel on sup Fig 4.*

We have added the percentage of the SDS gel to the figure legend of Supplementary Figure 4.

12. Do the authors have any explanations why CjCas9 did not work as base editor on targets that are cleaved efficiently by it?

Our previous study demonstrated that the efficiencies of Cas9-mediated base editing are usually lower than those of Cas9-mediated indel formation at the same target sites (Nishimasu *et al. Science* 2018), suggesting that more robust interactions between Cas9 and DNA are required to efficiently induce base editing as compared to indel formation. Although we lack a clear explanation for why CjCas9 does not work as a base editor, we speculate that enCjCas9, but not CjCas9, can bind a target DNA with sufficient affinity to induce base editing.

Reviewer #2:

Following up an earlier publication that characterized CjCas9, Nakagawa et al. re-analyzed PAM specificity using in vitro cleavage with a longer guide sequence (22mer) that previously used (20mer). The PAM is confirmed to be T3VRYAC. Authors then made an important effort in examining the capability of CjCas9 in genome and base-editing in human cells. A mutant variant, L58/K900 (enCjCas9), was also constructed based on its improved activity although with a slightly relaxed requirement for PAM position 7. In human cells, HEK293, CjCas9 and enCjCas9 showed high efficiency in inducing indels and base editing. This effort is a welcome advancement in genome editing field as CjCas9 is the smallest Cas9, which can improve portability. There are significant amount of data presented that support the conclusions. There are some suggestions that can improve the quality of the paper:

We thank the reviewer for the positive comments.

COMMENTS:

1. The efficiency demonstrated by CjCas9 (both wild-type and enhanced) in human cells is quite high and close to or better than that of SpCas9. Thus, this result is significant as many smaller and Type II-C Cas9 show decreased in vitro cleavage activities. It perhaps is important for

authors to discuss the process they used to optimize genome editing with CjCas9. Was there a major step/procedure that significantly improved the behavior of CjCas9 in HEK293 cells?

In this study, we found that it is critical to select 22-nt target sequences with optimal PAMs for successful CjCas9-mediated editing in HEK293 cells. Consistent with a previous study (Kim *et al.*, *Nat. Commun.* 2017), we confirmed that, while the widely used SpCas9 exhibits robust DNA cleavage activity with 20-nt guide sgRNAs, CjCas9 requires 22-nt guide sgRNAs for efficient DNA cleavage. Using the *in vitro* PAM discovery assay, we also found that CjCas9 has a previously unrecognized bias in N₃VRYAC PAMs and exhibits substantially reduced activities toward target sites with sub-optimal PAMs, such as N₃VGYAC, whereas previous studies reported that CjCas9 recognizes N₄RYAC PAMs (Kim *et al.*, *Nat. Commun.* 2017) or N₃VRYM PAMs (Yamada *et al.*, *Mol. Cell* 2017).

2. *The statement in discussion (Page 7, line 195) suggests that the target loci dependent activity of CjCas9 is a result of genome context. However, it appears that different targets are also associated with different PAM. For instance, the worst targets seem to be with N3AACAG (Figure 2a) rather than genome location. Thus, can authors re-qualify what they meant?*

Thank you for the helpful comments. CjCas9 exhibited similar trends for its PAM preference *in vitro* and *in vivo*, but displayed substantially different editing efficiencies at some target sites even with the same PAM sequences in human cells. For example, CjCas9 induced indels at the DYRK1A and MECP2 sites at >80% and <10% frequencies, respectively, although the two sites commonly have N₃GACAC PAMs, suggesting that the editing efficiencies are affected by the genomic context, as observed previously (Kim *et al.*, *Nat. Commun.* 2017). To avoid confusion, we have modified the statement in the Discussion of the revised manuscript.

3. *3) Page 5, line 50, The “V” designation is incorrect, it should be A/C/G not A/T/G. This created some confusion on description of the PAM specificities.*

Thank you for pointing it out. We changed “T” to “C” in the revised manuscript.

4. *Supplementary Figure 5 does not show the 40 CjCas9 variants as described. Furthermore, can authors explain their rationale for these 40 variants?*

Thank you for the helpful comment. We identified 14 residues close to the nucleic acids in the CjCas9–sgRNA–DNA complex structure. Thus, we purified over 40 variants (20 single mutants and their combinations), and measured their DNA cleavage activities *in vitro*. Since only two mutations (L58Y and D900K) substantially improved the DNA cleavage activity of CjCas9, we focused on the L58Y and D900K mutations in the original manuscript for clarity. According to the reviewer's suggestion, we have added a brief statement about the other mutations and showed the 14 mutated residues in Supplementary Figure 4a.

5. *There does not seem to have any information on the CjCas9 or CjCas9-base editor plasmids used for HEK293 transfection.*

We have added the detailed information about the plasmids used for *in vivo* editing experiments in Supplementary Table 3.

Reviewer #3:

The authors first determined the optimal guide RNA length by using an in vitro cleavage assay at a target sequence. Then, the authors also found the optimal PAM with another in vitro cleavage assay, again using only one target sequence. The results of both of these experiments are essentially in line with previous studies. Next, they developed enCjCas9, a variant of CjCas9, by introducing two mutations, L58Y and D900K. An in vitro cleavage assay showed that enCjCas9 has broader PAM compatibility than CjCas9. Then, the authors compared enCjCas9 with CjCas9 and SpCas9 in human cells. EnCjCas9 showed higher efficiencies than CjCas9 or SpCas9, although only two sites were used for the comparison with SpCas9. Next, the authors generated cytosine base editor by adding deaminase (PmCDA1) and UGI to CjCas9 D8A and enCjCas9 D8A nickases. Base editing was achieved only by enCjCas9 but its efficiencies were substantially lower than that of the SpCas9 nickase-based base editor:

We thank the reviewer for the positive comments.

MAJOR COMMENTS:

1. *The authors should test off-target effects of enCjCas9 using a minimum of two, or preferably three (or more), on-target sequences. We recommend using unbiased methods such as GUIDE-seq.*

Using the Digenome-seq method, Kim *et al.* identified the off-target sites of CjCas9-mediated DNA cleavage in human genomic DNA (Kim *et al.*, *Nat. Commun.* 2017). Accordingly, we examined the editing activities of CjCas9 and enCjCas9 toward two on-target sites (AAVS1-TS1 and AAVS1-TS8) and six reported off-target sites (AAVS1-TS1-02-04 and AAVS1-TS8-02-04). We found that CjCas9 and enCjCas9 efficiently edited the two on-target sites but not the six off-target sites, indicating that enCjCas9 exhibits higher on-target activity than CjCas9, but has off-target effects comparable to those of CjCas9. We have included these results in the revised manuscript (Supplementary Figure 6).

2. *Only two target sequences were used to compare the efficiencies of enCjCas9 and SpCas9. We recommend that the authors conduct the same experiments using more target sequences (preferably more than five) to draw more generalized conclusions supported by more compelling evidence.*

According to the reviewer's comments, we measured indel formations induced by CjCas9, enCjCas9 and SpCas9 at eight new target sites with NGGAACAC PAMs. Consistent with the results obtained with the two target sites tested in the original manuscript, CjCas9 and enCjCas9 induced indels at the eight sites at efficiencies comparable to or higher than those of SpCas9. We have included these results in the revised manuscript (Figure 2c).

3. *Can the enCjCas9-base editor sequence be packaged into one AAV vector? If not, the impact of enCjCas9 would be somewhat limited.*

Since the enCjCas9-AID construct is composed of 1,419 residues (4.3 kb), it cannot be packaged into one AAV vector. Nonetheless, CjCas9 fused with ABE8e, a recently reported compact deaminase (Richter *et al.*, *Nat. Biotechnol.* 2020), consists of 1,181 residues (3.5 kb) and can be

packaged, together with its sgRNA, into one AAV vector, whereas SpCas9-ABE8e consists of 1,565 residues (4.7 kb) and cannot be packaged into one AAV vector.

MINOR COMMENTS:

1. *“We purified more than 40 CjCas9 variants, and measured their cleavage activities toward the sub-optimal T3VGCAC targets (Supplementary Fig. 5b). Notably, the L58Y and D900K mutations enhanced the DNA cleavage activity, and the L58Y/D900K double mutation further improved the activity of CjCas9 (Supplementary Fig. 5c–f). Wrong figure numbers.*

We fixed the figure numbers in the revised manuscript.

2. *Line 146, there is a typo: sup-optimal*

We fixed it in the revised manuscript.

REVIEWERS' COMMENTS:

Reviewer #1 (Remarks to the Author):

In the revised manuscript, Nakagawa and colleagues addressed several of the reviewers' comments. With all due respect to Professors Nishimasu and Nureki, however, I must say that I am not particularly impressed with the revision of the MS.

No marked-up copy was provided, and the authors did not specify where in the MS changes were made, making it almost impossible to understand what was done with reasonable effort.

My impression is that they thanked me for the comments, stated that they had corrected the errors in the MS, but in fact did nothing substantive with several comments.

For example, my main concerns about the determination of the PAM sequence or the specificity of enCjCas9 were not addressed at all, except to thank me for my comments.

The specificity of enCjCas9 was addressed in the response of reviewer 3, but I would have expected a more comprehensive approach. I do not feel that the authors believe that their mutant Cas9 is worthy of a comprehensive characterization.

I still have problems with their rationale for determining the optimal spacer length in vitro as well. In the introduction they wrote:

"First, the optimal guide lengths for CjCas9 were determined in human cells (ref8) but not in vitro. ... These uncertainties might have hampered the wide use of CjCas9 as a versatile genome-editing tool."

I don't see that knowing the optimal in vitro spacer length has any impact on the use of CjCas9 in cellulo/in vivo, where the optimal spacer length is known and is not expected to be different anyway. In addition, although no statistics were used, the effect of 22 and 23 nucleotide spacer lengths does not appear to be different in Sup 2a, although the author inferred from the figure that 22 is the optimal length, giving the impression that they in fact also believe that this issue is not critical. They should give better rationale why they did these experiments.

The information they provide is still insufficient to understand exactly how the experiments were conducted, least of all to reproduce them.

The revised MS is still full of contradictions, to mention just a few:

What is the real length of the target used in the in vitro experiments?

"First, in vitro cleavage experiments were performed, using the purified CjCas9, four sgRNAs with different guide lengths (20–23-nt), and a linearized pUC plasmid containing the THE 23-NT TARGET SEQUENCE (complementary to the 20–23-nt guides) with the T3AACAC PAM (Supplementary Table 1). Next, the cleavage activities CjCas9 were measured, using the optimal 22-nt guide sgRNA and linearized pUC plasmids containing THE 23-NT TARGET SEQUENCE with 16 different PAMs."

However, in the next section:

"The PAM discovery assays were performed essentially as previously described using a library of pUC119 plasmids containing THE 22-NT TARGET SEQUENCE identical to that used in in vitro cleavage experiments." (I highlighted "THE 22/23-NT TARGET SEQUENCE" in capital letters)

So, is it 22 or 23?

Does the CjCas9 AID fit into an AAV vector or not?

In the Discussion:

"Given that the compact enCjCas9-AID (1,437 residues, 4.3 kb) CAN BE packaged into an adeno-associated virus vector, enCjCas9-AID would be useful for in vivo base editing"

In reply to the 3rd comment of reviewer 3:

"Since the enCjCas9-AID construct is composed of 1,419 residues (4.3 kb), IT CANNOT BE packaged into one AAV vector. Nonetheless, CjCas9 fused with ABE8e, a recently reported compact deaminase (Richter et al., Nat. Biotechnol. 2020), consists of 1,181 residues (3.5 kb) and can be packaged, together with its sgRNA, into one AAV vector, whereas SpCas9-ABE8e consists of 1,565 residues (4.7 kb) and cannot be packaged into one AAV vector."

It would be best to demonstrate this. Anyway, the use of hyperactive ABE8e is not necessarily a good idea due to its extremely high off-target propensity.

I apologize, but due to the lack of cooperation of the authors, I do not feel that a more detailed presentation of the weaknesses of the MS still present in the revised version would help to improve it. I still feel that enCjCas9 deserves a better presentation and more detailed characterization.

Reviewer #2 (Remarks to the Author):

Authors have now addressed the comments satisfactorily.

Reviewer #3 (Remarks to the Author):

The authors have addressed all of my concerns. I think that the manuscript is now suitable for publication.

Reviewer #1:

In the revised manuscript, Nakagawa and colleagues addressed several of the reviewers' comments. With all due respect to Professors Nishimasu and Nureki, however, I must say that I am not particularly impressed with the revision of the MS. No marked-up copy was provided, and the authors did not specify where in the MS changes were made, making it almost impossible to understand what was done with reasonable effort. My impression is that they thanked me for the comments, stated that they had corrected the errors in the MS, but in fact did nothing substantive with several comments.

We apologize for not specifying where we made changes in the previous version of our revised manuscript. We have shown the tracked changes in yellow in the revised manuscript.

The specificity of enCjCas9 was addressed in the response of reviewer 3, but I would have expected a more comprehensive approach. I do not feel that the authors believe that their mutant Cas9 is worthy of a comprehensive characterization.

As the reviewer suggested, GUIDE-seq is a comprehensive approach to detect Cas9-mediated off-target cleavage. However, GUIDE-seq is not easy to perform and has only been used by a limited number of laboratories. On the other hand, using Digenome-seq, another genome-wide off-target detection approach, Kim *et al.* already reported six off-target sites for CjCas9 (AAVS1-TS1-02-04 and AAVS1-TS8-02-04) in human genomic DNA (Kim *et al. Nat. Commun.* 2017). Thus, we consulted the editor and decided to examine the editing activities of CjCas9 and enCjCas9 toward these six known off-target sites, to investigate the specificity of enCas9.

I don't see that knowing the optimal in vitro spacer length has any impact on the use of CjCas9 in cellulo/in vivo, where the optimal spacer length is known and is not expected to be different anyway. In addition, although no statistics were used, the effect of 22 and 23 nucleotide spacer lengths does not appear to be different in Sup 2a, although the author inferred from the figure that 22 is the optimal length, giving the impression that they in fact also believe that this issue is not critical. They should give better rational why they did these experiments.

We agree with the reviewer that the optimal guide (spacer) length is essentially the same *in vivo* and in cells. Indeed, our *in vitro* cleavage experiments revealed that the optimal guide length for

CjCas9 is 22-nucleotides, consistent with the previous study showing that CjCas9 exhibits the highest editing activities with sgRNAs with a 22-nucleotide guide in human cells (Kim *et al. Nat. Commun.* 2017). Nonetheless, *in vivo* genome-editing experiments do not enable direct measurements of Cas9-mediated DNA cleavage, and instead evaluate indel formation after Cas9-induced double-strand breaks, which could be affected by many factors, such as genomic contexts. In contrast, *in vitro* DNA cleavage experiments enable more accurate measurements of Cas9-mediated DNA cleavage reactions. Therefore, we believe that it is important to determine the optimal guide length by *in vitro* DNA cleavage experiments for Cas9-mediated genome-editing applications.

The information they provide is still insufficient to understand exactly how the experiments were conducted, least of all to reproduce them. The revised MS is still full of contradictions, to mention just a few: What is the real length of the target used in the in vitro experiments? "First, in vitro cleavage experiments were performed, using the purified CjCas9, four sgRNAs with different guide lengths (20–23-nt), and a linearized pUC plasmid containing the THE 23-NT TARGET SEQUENCE (complementary to the 20–23-nt guides) with the T3AACAC PAM (Supplementary Table 1). Next, the cleavage activities CjCas9 were measured, using the optimal 22-nt guide sgRNA and linearized pUC plasmids containing THE 23-NT TARGET SEQUENCE with 16 different PAMs." However, in the next section: "The PAM discovery assays were performed essentially as previously described using a library of pUC119 plasmids containing THE 22-NT TARGET SEQUENCE identical to that used in in vitro cleavage experiments." (I highlighted "THE 22/23-NT TARGET SEQUENCE" in capital letters)

For all *in vitro* cleavage assays, we used pUC plasmids containing the common 23-nt target sequence (GGGGGAAATTAGGTGCGCTTGGC, the 22-nt target sequence is underlined) and different PAM sequences (Supplementary Table 1). To optimize the guide length, we used four sgRNAs with different guide lengths (sgRNA-20, GGAAATTAGGTGCGCTTGGC; sgRNA-21, GGGAAATTAGGTGCGCTTGGC; sgRNA-22, GGGGGAAATTAGGTGCGCTTGGC; sgRNA-23, GGGGGAAATTAGGTGCGCTTGGC) and a linearized pUC plasmid containing the T₃AACAC PAM and the 23-nt target sequence (GGGGGAAATTAGGTGCGCTTGGC), which is thus complementary to all of the sgRNAs (sgRNA-20, sgRNA-21, sgRNA-22, and sgRNA-23). To examine the PAM specificity, we used the sgRNA with the optimal 22-nt guide (GGGGGAAATTAGGTGCGCTTGGC) and linearized pUC plasmids containing the same 23-nt target sequence

(GGGGGAAATTAGGTGCGCTTGGC) with 16 different PAMs. For the PAM discovery assay, we used the sgRNA with the optimal 22-nt guide (GGGGGAAATTAGGTGCGCTTGGC) and a library of pUC119 plasmids containing the same 23-nt target sequence (GGGGGAAATTAGGTGCGCTTGGC) with adjacent randomized 8-bp sequences. To avoid confusion, we have modified the statements in the Results and Methods sections in the revised manuscript.

Does the CjCas9 AID fit into an AAV vector or not? In the Discussion: “Given that the compact enCjCas9-AID (1,437 residues, 4.3 kb) CAN BE packaged into an adeno-associated virus vector, enCjCas9-AID would be useful for in vivo base editing” In reply to the 3rd comment of reviewer 3: “Since the enCjCas9-AID construct is composed of 1,419 residues (4.3 kb), IT CANNOT BE packaged into one AAV vector. Nonetheless, CjCas9 fused with ABE8e, a recently reported compact deaminase (Richter et al., Nat. Biotechnol. 2020), consists of 1,181 residues (3.5 kb) and can be packaged, together with its sgRNA, into one AAV vector, whereas SpCas9-ABE8e consists of 1,565 residues (4.7 kb) and cannot be packaged into one AAV vector.” It would be best to demonstrate this. Anyway, the use of hyperactive ABE8e is not necessarily a good idea due to its extremely high off-target propensity.

Thank you for your critical comments. As the reviewer pointed out, without experimental evidence, we cannot explicitly state that enCjCas9-AID (4.3 kb) can be packaged with its sgRNA (0.1 kb) into a single AAV vector, although AAV generally has a packaging capacity of 4.5–4.7 kb. Thus, we have modified the statement “Given that the compact enCjCas9-AID (1,437 residues, 4.3 kb) can be packaged into an adeno-associated virus vector, enCjCas9-AID would be useful for *in vivo* base editing, thus expanding the utility of the CRISPR-Cas toolbox.” to “It is possible that the compact enCjCas9-AID (1,437 residues, 4.3 kb) can be packaged into an adeno-associated virus vector, thereby facilitating *in vivo* base editing”, in the revised manuscript.

Reviewer #2

Authors have now addressed the comments satisfactorily.

Thank you for the positive comments for our revised manuscript.

Reviewer #3

The authors have addressed all of my concerns. I think that the manuscript is now suitable for publication.

Thank you for the positive comments for our revised manuscript.